# FroSSL: Frobenius Norm Minimization for Self-Supervised Learning

## Abstract

Self-supervised learning (SSL) is an increasingly popular paradigm for representation learning. Recent methods can be classified as sample-contrastive, dimension-contrastive, or asymmetric network-based, with each family having its own approach to avoiding informational collapse. While dimension-contrastive methods converge to similar solutions as sample-contrastive methods, it can be empirically shown that some methods require more epochs of training to converge. Motivated by closing this divide, we present the objective function FroSSL which is both sample- and dimension-contrastive up to embedding normalization. FroSSL works by minimizing covariance Frobenius norms for avoiding collapse and minimizing mean-squared error for augmentation invariance. We show that FroSSL converges more quickly than a variety of other SSL methods and provide theoretical and empirical support that this faster convergence is due to how FroSSL affects the eigenvalues of the embedding covariance matrices. We also show that FroSSL learns competitive representations on linear probe evaluation when used to train a ResNet18 on the CIFAR-10, CIFAR-100, STL-10, and ImageNet datasets.

## 1 Introduction

The problem of learning representations without human supervision is fundamental in machine learning. Unsupervised representation learning is particularly useful when label information is difficult to obtain or noisy. It requires the identification of structure in data without any preconceptions about what the structure is. One common way of learning structure without labels is self-supervised learning (SSL). Recently, a flurry of SSL approaches have been proposed for learning visual representations (Chen et al., 2020a; HaoChen et al., 2021; Tsai et al., 2021b; Chen & He, 2021; Grill et al., 2020; He et al., 2020; Zbontar et al., 2021; Li et al., 2021). The basic goal of SSL is to train neural networks to capture *semantic* input features that are *augmentation-invariant*. This goal is appealing for representation learning because the inference set often has similar semantic content to the training set. We provide a more rigorous definition of this process in Section 2.1.

A trivial solution to learning augmentation-invariant features is to learn networks that encode *every* image to the same point. Such a solution is known as informational collapse and is of course useless for downstream tasks. SSL approaches can be roughly divided into three families, each with its own method of avoiding collapse. The first family consists of **sample-contrastive** methods (Chen et al., 2020a; HaoChen et al., 2021; Tsai et al., 2021b; He et al., 2020; Caron et al., 2020) which use $Z_{1,i}$ and $Z_{2,i}$ as positive samples and all $Z_{1,j}, Z_{2,j}, i \neq j$ as negative samples. Here $Z_1$ and $Z_2$ are the embeddings of views 1 and 2, as shown in Figure 1. Sample-contrastive methods use a contrastive loss to explicitly bring the positive samples close together while pushing the negative samples apart. The second family consists of **asymmetric network** methods (Chen & He, 2021; Grill et al., 2020; Caron et al., 2021) which place restrictions on the network architectures used. Restrictions include stop gradients as in Chen & He (2021) and asymmetrical encoders as in Grill et al. (2020). Interestingly, the objective functions typically used by this family allow for collapse, though this is avoided in practice due to the architectural restrictions. The third, and most recent, family are the **dimension-contrastive** methods (Zbontar et al., 2021; Bardes et al., 2022; Ermolov et al., 2021). These methods operate by reducing the redundancy in feature dimensions. Methods in this family are able to avoid the use of negative samples while also not requiring restrictions in the network architecture to prevent collapse.

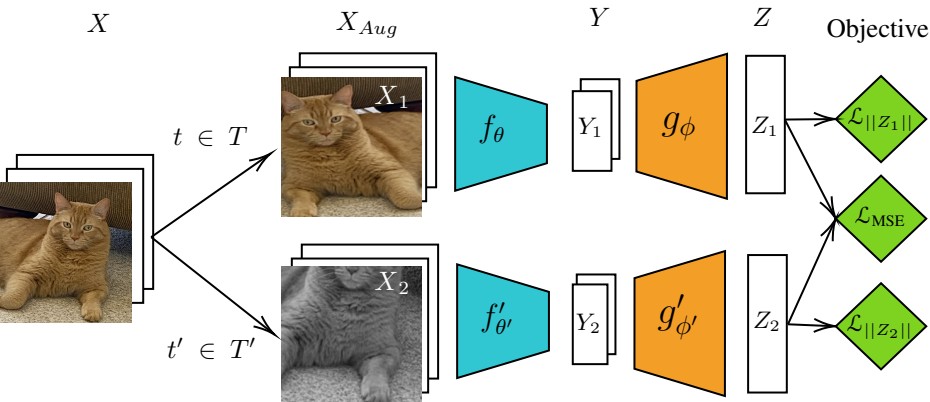

Figure 1: The SSL pipeline used in this work. In general, the encoder and projector may be asymmetric. We use symmetric encoders with shared weights and the same augmentation set for each view. We refer to $X_1$ as view 1 of $X$, and $X_2$ as view 2.

One disadvantage common to all current SSL methods is their speed of convergence. When compared to traditional supervised learning, SSL methods must be trained for large numbers of iterations to reach convergence. For example, a typical experiment in the literature is to train for 1000 epochs on ImageNet which can take several weeks even with 4 GPUs. An imperative direction of research is to investigate how to reduce SSL training time. An observation that is often hidden by only reporting the final epoch accuracy is that, empirically, certain SSL methods seem to converge slower than others. This phenomenon has been observed in Simon et al. (2023) but not discussed in detail. We provide additional support for this claim in Section 5. Our work attempts to answer the following research question: Does there exist an SSL method with dimension-contrastive advantages, namely simplicity via avoidance of both negative sampling and architectural restrictions, while achieving a superior speed of convergence to other existing SSL methods?

We propose an SSL objective which we call FroSSL. Similar to many dimension-contrastive methods, FroSSL consists of a variance and invariance term. The invariance term is simply a mean-squared error between the views and is identical to VICReg's invariance term (Bardes et al., 2022). The variance term is the log of the squared Frobenius norm of the normalized covariance embedding matrices. To the best of our knowledge, using the Frobenius norm of covariance matrices has not been explored in SSL. Our contribution can be summarized as:

- We introduce the FroSSL objective function and show that it is *both* dimension-contrastive and sample-contrastive up to a normalization of the embeddings.

- We evaluate FroSSL on the standard setup of SSL pretraining and linear probe evaluation on CIFAR-10, CIFAR-100, STL-10, and Imagenet. We find that FroSSL achieves strong performance, especially when models are trained for fewer epochs.

- We examine the covariance eigenvalues of various SSL methods to show which methods lead to the best-conditioned, and thus quickest, optimization problems.

## 2 BACKGROUND

Consider a matrix $A \in \mathbb{R}^{m \times n}$. Let $A_{ij} \in \mathbb{R}$ be the element at the $i$-th row and $j$-th column of $A$. Let $A_{i,:} \in \mathbb{R}^m$ be a column vector representing the $i$-th row of $A$. Let $\sigma_k(A)$ be the $k$-th largest singular value of $A$. If $A$ is square, let $\lambda_k(A)$ the $k$-th largest eigenvalue of $A$. An elementwise exponent is denoted as $A^{\odot p}$, while an element-wise product (Hadamard product) is denoted as $A \odot B$. The Frobenius norm of $A$ is defined as:

$$||A||_F^2 = \sum_i^m \sum_j^n A_{ij}^2 = \sum_k^{\min(m,n)} \sigma_k^2(A). \tag{1}$$

Table 1: Taxonomy of dimension-contrastive SSL methods describing how they avoid informational collapse and achieve augmentation invariance

| Method | Variance | Invariance |
|---|---|---|
| Barlow Twins | Cross-correlation off-diagonals | Cross-correlation diagonals |
| VICReg | (Variance) Hinge loss on auto-covariance diagonal | MSE |
| | (Covariance) covariance off-diagonals per view | |
| W-MSE | Implicit through whitening | MSE |
| CorInfoMax | Log-determinant entropy of covariance per view | MSE |
| FroSSL (ours) | Log of normalized covariance Frobenius norm per view | MSE |

For any real matrix $A$, we have:

$$||A^T A||_F = ||AA^T||_F \tag{2}$$

## 2.1 THE SELF-SUPERVISED LEARNING PROBLEM

Many visual SSL methods follow a similar procedure which was first introduced in Chen et al. (2020a). An example of this procedure is depicted in Figure 1. Let $\mathbf{X} = \{x_i\}_{i=1}^n$ be a mini-batch with $n$ samples. Let $T(\cdot)$ be a function that applies a randomly selected transformation to an image from a set of image transformations (augmentations). Let $f$ be a visual encoder network and $g$ be a projector network.

First, each image $x_i \in \mathbf{X}$ is paired with augmented versions of itself, making the augmented dataset $\mathbf{X}_{\text{aug}} = \{T(x_i), T(x_i)\}_{i=1}^n = \{X_1, X_2\}$[1]. Note that $X_{1,i}$ and $X_{2,i}$ have identical *semantic* content, but different *style* content. Second, this paired augmented dataset is passed through the networks to get $d$-dimensional embeddings $\mathbf{Z} = g(f(\mathbf{X}_{\text{aug}})) = Z_1, Z_2$. Finally, an SSL objective is computed on the embeddings and backpropagated through both networks. The goal of the objective is to ensure that the paired images are mapped close together, i.e. $Z_{1,i} \approx Z_{2,i}$. Thus the goal of SSL is to train the networks to extract *semantic* features that are invariant to any augmentations that can be computed using $T(\cdot)$.

## 2.2 DIMENSION-CONTRASTIVE METHODS

The dimension-contrastive methods, which are sometimes called negative-free contrastive (Tsai et al., 2021a) or feature decorrelation methods (Tao et al., 2022), operate by reducing the redundancy in feature dimensions. Instead of examining *where* samples live in feature space, these methods examine *how* feature dimensions are being used.

Many recent works in dimension-contrastive SSL, whether explicitly or implicitly, consist of having a loss function that fulfills two roles:

- **Variance** This is the *explosion factor* that ensures informational collapse is avoided.

- **Invariance** This is the *implosion factor* that ensures useful augmentation-invariant representations are learned.

SSL methods belonging to the dimension-contrastive family include Barlow Twins (Zbontar et al., 2021), VICReg (Bardes et al., 2022), W-MSE (Ermolov et al., 2021), and CorInfoMax (Ozsoy et al., 2022). Barlow Twins objective pushes the normalized cross-covariance between views towards the identity matrix. VICReg consists of three terms, dubbed variance, invariance, and covariance. The invariance term enforces similarity in embeddings across views, while the variance/covariance terms regularize the covariance matrices of each view to prevent collapse. W-MSE whitens and projects embeddings to the unit sphere before maximizing cosine similarity between positive samples. Finally, CorInfoMax maximizes the $\log \det$ entropy of both views while minimizing mean-squared error. A taxonomy of these methods is shown in Table 1.

---

[1] $\{T(x_i), T(x_i)\}$ should be understood as making to separate calls to the function $T$. For each call a transformation is selected at random

## 3 THE FROSSL OBJECTIVE

To motivate FroSSL, we begin by examining the Barlow Twins objective,

$$\mathcal{L}_{\text{Barlow}} = \sum_i (1 - M_{ii})^2 + \lambda \sum_i \sum_{i \neq j} {M_{ij}}^2 \tag{3}$$

where $M$ is the cross-correlation matrix. Without feature normalization, the objective $\mathcal{L}_{\text{Barlow}}$ pushes $M$ to approach identity and is not rotationally invariant. However, we posit that dimension-contrastive methods *should* be rotationally invariant because the orientation of the covariance does not affect the relationships between principal components. In other words, redundancy in the embedding dimensions is invariant to the rotation of the embeddings. Thus dimension-contrastive methods should be rotationally invariant as well.

One natural matrix operation that is invariant to unitary transformations is the Frobenius norm. Minimizing the Frobenius norm of normalized embeddings will cause the embeddings to spread out equally in all directions. Normalizing the embeddings is crucial because otherwise, minimizing the Frobenius norm will lead to trivial collapse. We propose to use the following term to reduce redundancy between dimensions:

$$\mathcal{L}_{\text{Fro}} = \log(||Z_1^T Z_1||_F^2) + \log(||Z_2^T Z_2||_F^2) \tag{4}$$

The $\mathcal{L}_{\text{Fro}}$ fills the role of a variance term. For the invariance term, we can simply use mean-squared error between the views, defined as

$$\mathcal{L}_{\text{MSE}} = \frac{1}{n} \sum_{i=1}^n ||z_{1,i} - z_{2,i}||_2^2 \tag{5}$$

Combining (4) and (5) yields the FroSSL objective.

$$\text{minimize } \mathcal{L}_{\text{FroSSL}} = \log(||Z_1^T Z_1||_F^2) + \log(||Z_2^T Z_2||_F^2) + \frac{1}{N} \sum_{i=1}^n ||z_{1,i} - z_{2,i}||_2^2 \tag{6}$$

Due to Equation (2), we can choose to compute either $||Z_1^T Z_1||_F^2$ or $||Z_1 Z_1^T||_F$ depending on if $d > n$. The former has time complexity $O(nd^2)$ while the latter has complexity $O(n^2 d)$. For consistency, we always use the former in our experiments. We provide Pytorch-style pseudocode in Appendix A.

### 3.1 THE ROLE OF THE LOGARITHM

The role of the logarithms in (4) is twofold. First, the logarithm allows interpreting $\mathcal{L}_{\text{Fro}}$ as entropy maximization. One recent information-theoretic framework with success in deep learning is matrix-based entropy (Sanchez Giraldo et al., 2015). It is an information-theoretic quantity that behaves similarly to Rényi's $\alpha$-order entropy, but it can be estimated directly from data without making strong assumptions about the underlying distribution. In particular, the first and second terms of (4) correspond to the matrix-based negative collision entropies of $Z_1$ and $Z_2$. This is relevant because collision entropy measures the coincidence of points in a space. By maximizing collision entropy, the coincidence of points is minimized and trivial collapse is avoided.

Second, we hypothesize that the $\log$ ensures that the contributions of the variance term to the gradient of the objective function become self regulated ($\frac{d \log f(x)}{dx} = \frac{1}{f(x)} \frac{df(x)}{dx}$) with respect to the invariance term. Initially we attempted using tradeoffs between (4) and (5). However, a grid search showed that the optimal tradeoff was when the terms were equally weighted. This is a nice advantage over methods such as Barlow Twins and VICReg, where the choice of tradeoff hyperparameters is crucial to the performance of the model. We later compare the experimental performance of Equation (6) with and without the logarithms, showing that using logarithms leads to a gain in performance.

### 3.2 FROSSL IS BOTH SAMPLE-CONTRASTIVE AND DIMENSION-CONTRASTIVE

It can be shown, up to an embedding normalization, that FroSSL is both dimension-contrastive and sample-contrastive. First, we provide formal definitions of dimension-contrastive and sample-contrastive SSL that were first proposed in Garrido et al. (2023b).

**Definition 3.1** (Dimension-Contrastive Method)**.** An SSL method is said to be dimension-contrastive if it minimizes the non-contrastive criterion $\mathcal{L}_{nc}(Z) = ||Z^T Z - \text{diag}(Z^T Z)||_F^2$, where $Z \in \mathbb{R}^{N \times D}$ is a matrix of embeddings as defined above. This may be interpreted as penalizing the off-diagonal terms of the embedding covariance.

**Definition 3.2** (Sample-Contrastive Method)**.** An SSL method is said to be sample-contrastive if it minimizes the contrastive criterion $\mathcal{L}_c(Z) = ||ZZ^T - \text{diag}(ZZ^T)||_F^2$. This may be interpreted as penalizing the similarity between pairs of different images.

Next, we use the duality of the Frobenius norm, as shown in Equation (2), to show that FroSSL satisfies the qualifying criteria of both dimension-contrastive and sample-contrastive methods.

**Proposition 3.1.** If every embedding dimension is normalized to have equal variance, then FroSSL is a dimension-contrastive method. The proof is shown in Appendix E.1.

**Proposition 3.2.** If every embedding is normalized to have equal norm, then FroSSL is a sample-contrastive method. The proof is shown in Appendix E.2.

**Proposition 3.3.** If the embedding matrices are doubly stochastic, then FroSSL is simultaneously dimension-contrastive and sample-contrastive.

Proposition 3.3 allows for interpreting FroSSL as either a sample-contrastive or dimension-contrastive method, up to a normalization of the data embeddings. The choice of normalization strategy is not of particular importance to the performance of an SSL method (Garrido et al., 2023b). Unless otherwise specified, we only normalize the variance and not the embeddings. Another method that shares these properties is TiCo (Zhu et al., 2022). Additionally, variants of the dimension-contrastive VICReg were introduced in Garrido et al. (2023b) that allowed it to be rewritten as the sample-contrastive SimCLR. However, VICReg itself is not able to be rewritten in such a way.

## 4 RELATED WORK

### 4.1 EXISTING SSL METHODS

The dimension-contrastive family of SSL methods was discussed in Section 2.2. The sample-contrastive family of methods operates by discriminating positive and negative pairs of samples. Many sample-contrastive methods require large batch sizes for the best performance, However, this is not a property that FroSSL shares. Prominent methods in this family include SimCLR (Chen et al., 2020a), MoCo (He et al., 2020; Chen et al., 2020b), and SwAV (Caron et al., 2020). SimCLR first introduced projector heads and data augmentation for positive sample generation, both of which have become prevalent in the SSL literature. MoCo built upon SimCLR and introduced momentum encoders, which improved training stability, as well as a memory bank to mitigate the large batch size requirements of SimCLR. On the other hand, SwAV relaxed the sample discrimination problem by instead contrasting cluster assignments. SwAV was shown to perform well even with small batch sizes without requiring a momentum encoder or memory bank.

The asymmetric network methods employ a variety of architectural techniques in order to prevent trivial collapse. These techniques include asymmetrical encoders (Chen & He, 2021; Grill et al., 2020), momentum encoders (He et al., 2020), and stop gradients (Chen & He, 2021). While these methods can achieve great results, they are rooted in implementation details and there is no clear theoretical understanding of how they avoid collapse (Bardes et al., 2022).

### 4.2 SSL METHODS USING KERNELS

There is prior work in SSL that uses kernel-based objectives for learning representations, much like we do. SSL-HSIC (Li et al., 2021) uses an objective based on the Hilbert-Schmidt Independence Criterion (Gretton et al., 2007), which itself has ties to matrix-based entropy. TiCo (Zhu et al., 2022) considers the theoretical connections between kernel Gram matrices and covariance matrices. TiCo also makes use of an exponential moving average on covariance matrices which serves as a memory bank.

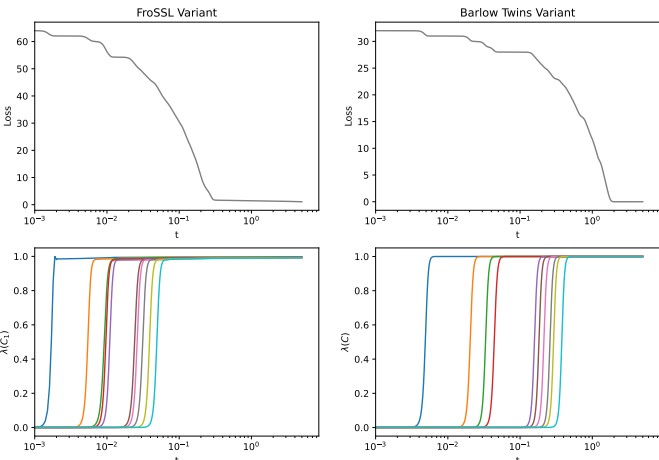

Figure 2: A side-by-side comparison of the FroSSL variant in (8) and the Barlow Twins variant from Simon et al. (2023). The top row shows the loss and the bottom row shows the top 10 eigenvalues of the View 1 covariance matrix. The x-axis is t = lr * step.

### 4.3 ENTROPY IN SSL

The FroSSL objective is closely related to the CorInfoMax objective proposed in Ozsoy et al. (2022).

$$\max \mathcal{L}_{\text{CorInfoMax}} = \log \det(Z_1^T Z_1 + \epsilon I) + \log \det(Z_2^T Z_2 + \epsilon I) - \beta \mathcal{L}_{\text{MSE}} \tag{7}$$

The CorInfoMax objective uses $\log \det$ entropy, as opposed to the matrix-based entropy described in Section 3.1. One advantage of our approach is that the Frobenius norm can be computed in $O(d^2 n)$, assuming that $d < n$. On the other hand, $\log \det$ entropy always requires computing the eigendecomposition which is $O(d^3)$. Another advantage of FroSSL over CorInfoMax is the absence of hyperparameters in the objective. We found the selection of $\epsilon$ to be critical for avoiding instabilities in the eigendecomposition.

Another recent work that uses entropy is SimMER (Yang et al., 2022). Rather than $\log \det$ or matrix-based entropy, SimMER uses an entropy estimator based on nearest neighbors (Kozachenko & Leonenko, 1987). SimMER is not negative-free because the estimator implicitly chooses the nearest neighboring point as a negative. We hypothesize that using matrix-based entropy, via the Frobenius norm, instead of nearest-neighbor entropy estimators allows for more robust representations.

## 5 THE TRAINING DYNAMICS OF FROSSL

### 5.1 STEPWISE CONVERGENCE IN THE LINEAR REGIME

Recent work has examined the training dynamics of SSL models (Simon et al., 2023). In particular, they find that the eigenvalues of the covariance exhibit "stepwise" behavior, meaning that one eigendirection is learned at a time. They claim that this phenomenon contributes to slowness in SSL optimization because the smallest eigendirections take the longest to be learned. This is supported by a recent finding that shows that high-rank representations lead to better classification accuracies (Garrido et al., 2023a). An interesting line of analysis shown in Simon et al. (2023) is provable stepwise convergence with linear networks. Linear networks are appealing theoretical tools because one can work out what they converge to. Inspired by (Simon et al., 2023; Garrido et al., 2023b; Balestriero & LeCun, 2022), we introduce a slightly simplified variant of FroSSL which is amenable to analysis in the linear regime:

$$\mathcal{L} = ||Z_1^T Z_1 - I_d||_F^2 + ||Z_2^T Z_2 - I_d||_F^2 + ||Z_1 - Z_2||_F^2 \tag{8}$$

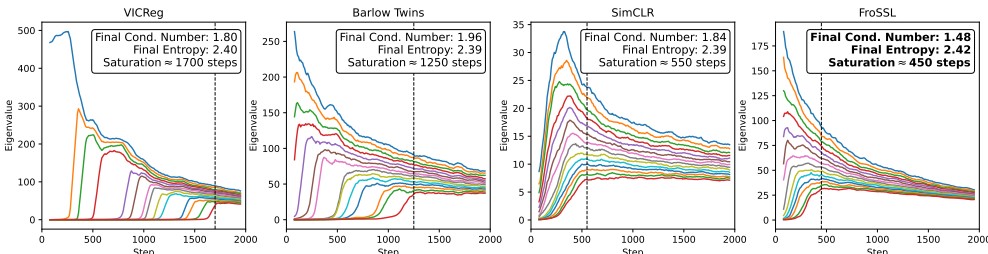

Figure 3: The top 14 eigenvalues of the embedding covariance $Z_1^T Z_1$. The condition number and eigenvalue Shannon entropy are shown for the end of epoch 5 (roughly 2000 steps). A vertical line marks the saturation of the $14^{\text{th}}$ eigenvalue. The best quantities are **bolded**.

While not included in the main text, we work out exact training dynamics in Appendix D. In particular, we show the optimal representation and a closed-form solution for the linear layer at each training step.

Shown in Figure 2 is a comparison between Equation (8) and the Barlow Twins variant $||Z_1^T Z_2 - I_d||_F^2$ studied in Simon et al. (2023). We train two linear layers, one for each view, using full batch gradient descent on 1024 samples drawn from CIFAR10. It is readily observed that (8) converges much quicker.

## 5.2 STEPWISE CONVERGENCE IN THE NONLINEAR REGIME

The phenomenon of stepwise convergence occurs in the nonlinear regime as well. We create an experimental setup similar to the one used in Simon et al. (2023). For all SSL objectives, a ResNet18 was trained on STL10 using $lr = 0.1$ and a batch size of 256. The learning rate was chosen by performing a sweep over {1e-4, 1e-3, 1e-2, 1e-1} and selecting the one that led to the highest linear probe accuracy after 100 epochs. A learning rate of 0.1 was best for all objectives. Further experimental details are given in B.1.

In Figure 3, we compare FroSSL to VICReg, Barlow Twins, and SimCLR. We train for 5 epochs and plot the top 14 eigenvalues of the view 1 covariance $Z_1^T Z_1$ over time. At the end of the 5th epoch, FroSSL outperforms the other methods in the following three metrics:

- **Condition Number** Given by $\frac{\lambda_1(Z_1^T Z_1)}{\lambda_{14}(Z_1^T Z_1)}$. The ideal condition number is 1 because the smallest eigendirection is as relevant as the largest.
- **Shannon Entropy** Given by $-\sum_i \lambda_i \log(\lambda_i)$, where the eigenvalues are normalized to sum to 1 before computation. The optimal value here is maximum entropy, which is obtained when all eigenvalues are equal. Higher entropy is better because more eigendirections have been learned.
- **Saturation** Given by the step at which the 14th eigenvalue saturates. Earlier is better because convergence can occur with fewer training steps.

We speculate that FroSSL allows the covariance eigenvalues to converge quicker because per Equation (1), the $\mathcal{L}_{\text{Fro}}$ can be rewritten as below. This shows that if the embedding dimensions are normalized to have variance $\rho$, then $\mathcal{L}_{\text{Fro}}$ explicitly tries to make the covariance eigenvalues approach to $\rho$.

$$\mathcal{L}_{\text{Fro}} = \log(||Z_1^T Z_1||_F^2) + \log(||Z_2^T Z_2||_F^2) = \log\left(\sum_i^d \lambda_i^2(Z_1^T Z_1)\right) + \log\left(\sum_i^d \lambda_i^2(Z_2^T Z_2)\right) \quad (9)$$

## 6 EXPERIMENTAL RESULTS

We use a standard linear probe evaluation protocol, which is pretraining a ResNet18 backbone and then training a linear classifier on the representation, on the CIFAR-10, CIFAR-100, STL-10, and

Table 2: Comparison of SSL methods on small datasets. CIFAR-10 and CIFAR-100 were trained for 1000 epochs with baseline results reported from da Costa et al. (2022); Ermolov et al. (2021). STL-10 was trained for 500 epochs and all baseline results are from our implementation. Best result is in **bold**, second best is underlined.

| Method | CIFAR-10 | CIFAR-100 | STL-10 | Average |
|---|---|---|---|---|
| **Sample-Contrastive** | | | | |
| SimCLR | 91.8 | 65.8 | 85.9 | 81.2 |
| SwAV | 89.2 | 64.9 | 82.6 | 78.9 |
| MoCo v2 | **92.9** | 69.9 | 83.2 | 82.0 |
| **Asymmetric Network** | | | | |
| SimSiam | 90.5 | 66.0 | 88.5 | 81.7 |
| BYOL | 92.6 | 70.5 | **88.7** | **83.9** |
| DINO | 89.5 | 66.8 | 78.9 | 78.4 |
| **Dimension-Contrastive** | | | | |
| VICReg | 92.1 | 68.5 | 85.9 | 82.2 |
| Barlow Twins | 92.1 | **70.9** | 85.0 | 82.7 |
| W-MSE 2 | 91.6 | 66.1 | 72.4 | 76.7 |
| CorInfoMax | 92.6 | 69.7 | - | - |
| FroSSL (no logs) | 88.9 | 62.3 | 82.4 | 77.9 |
| FroSSL | 92.8 | 70.6 | 87.3 | 83.6 |

ImageNet datasets. The first three datasets are presented in Section 6.1. while the latter is shown in Section 6.2.

## 6.1 EVALUATION ON SMALL DATASETS

For CIFAR-10, CIFAR-100, and STL-10, we use the solo-learn SSL framework (da Costa et al., 2022). In Table 2, we show linear probe evaluation results on these datasets. It is readily seen that FroSSL learns competitive representations with other SSL methods. For methods other than FroSSL and CorInfoMax, we show CIFAR-10 and CIFAR-100 results from da Costa et al. (2022); Ermolov et al. (2021). In our experience, CorInfoMax is sensitive to choice of hyperparameters and we were not able to get it to converge on STL-10. The implementation details can be summarized as:

- **Optimizer** The backbone uses LARS optimizer (You et al., 2017) with an initial learning rate of 0.3, weight decay of 1e-6, and a warmup cosine learning rate scheduler. The linear probe uses the SGD optimizer (Kingma & Ba, 2014) with an initial learning rate of 0.3, no weight decay, and a step learning rate scheduler with decreases at 60 and 80 epochs.

- **Epochs** For CIFAR-10 and CIFAR-100, we pretrain the backbone for 1000 epochs. For STL-10, we pretrain for 500 epochs. All linear probes were trained for 100 epochs.

- **Hardware** The backbones were trained on one NVIDIA V100 GPU.

- **Hyperparameters** For methods other than FroSSL, we use the CIFAR-100 hyperparameters reported in da Costa et al. (2022) on the STL-10 dataset. A batch size of 256 is used for all methods.

In Table 3, online linear classifier accuracies are shown for STL-10 on several epochs during training. FroSSL outperforms all other dimension-contrastive methods. Another observation is that for the first 30 epochs, FroSSL outperforms *all* other SSL methods shown. This trend complements the empirical stepwise convergence results discussed in Section 5.2. In the subsequent section, we will see if this trend scales up to ImageNet.

## 6.2 EVALUATION ON IMAGENET

Here we use FroSSL to train a ResNet18 on ImageNet for 100 epochs. We compare to Barlow Twins on the exact same setup. We show the top1 and top5 accuracies in the first 30 epochs in Figure 4. Even after the first epoch, FroSSL has an improvement of 12.2% over Barlow Twins. We show the first 30 epochs to emphasize what happens early in training. Afterward, Barlow Twins does catch up to FroSSL and achieves similar performances. FroSSL and Barlow Twins achieve final top1/top5 accuracies of 53.4/77.7 and 52.5/77.5. The implementation details can be summarized as:

- **Optimizer** The backbone uses stochastic gradient descent (SGD) with an initial learning rate of 1e-2, weight decay of 5e-4, and a cosine annealing scheduler with warm restarts

Table 3: Top-1 Accuracies on STL-10 using an online linear classifier during training.

| | Epoch | | | | |
| Method | 3 | 10 | 30 | 50 | 100 |
| --- | --- | --- | --- | --- | --- |
| **Sample-Contrastive** | | | | | |
| SimCLR | 40.7 | 44.8 | 61.5 | 66.2 | 70.1 |
| SwAV | 30.9 | 38.7 | 64.6 | 69.3 | 74.3 |
| MoCo v2 | 24.6 | 45.0 | 63.8 | **69.4** | 75.2 |
| **Asymmetric Networks** | | | | | |
| SimSiam | 31.8 | 41.2 | 54.7 | 65.6 | **77.1** |
| BYOL | 28.8 | 32.7 | 59.6 | 64.7 | 70.6 |
| DINO | 26.6 | 26.7 | 38.2 | 43.2 | 46.1 |
| **Dimension-Contrastive** | | | | | |
| VICReg | 43.6 | 51.1 | 61.2 | 67.5 | 71.1 |
| Barlow Twins | 32.1 | 46.6 | 62.0 | 62.6 | 69.0 |
| W-MSE 2 | 17.2 | 30.4 | 45.6 | 53.4 | 61.9 |
| FroSSL (no logs) | 40.5 | 51.9 | 60.6 | 64.1 | 67.3 |
| FroSSL | **44.8** | **56.9** | **64.8** | 67.1 | 72.0 |

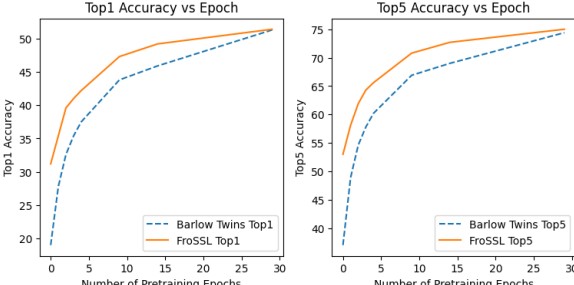

Figure 4: Comparison of SSL methods when training a ResNet18 on ImageNet.

every 15 epochs. The linear probe uses the Adam optimizer with an initial learning rate of 5e-3, no weight decay, and a step learning rate scheduler with decreases every 10 epochs.

- **Epochs** The backbone is trained for 100 epochs. Linear probes were trained for 100 epochs.

- **Hardware** The backbones were trained on 4 NVIDIA A100 (40GB) GPUs.

- **Hyperparameters** We use $\lambda$=5e-3 for Barlow Twins as recommended in Zbontar et al. (2021). An effective batch size of 224 was used for the backbones, which equates to 56 samples per GPU. We use the same augmentation set as Chen et al. (2020a).

### 6.3 ABLATIONS

In Tables 2 and 3, we test a variant of FroSSL with no logarithms. This variant has obviously worse performance than FroSSL. Importantly, we do not use any tradeoff hyperparameter between the invariance and variance terms. While such a hyperparameter may improve performance, one intuition in Section 3.1 was that the logarithm acts as a natural alternative to tradeoffs. Furthermore, simply adding a logarithm to an objective function is more straightforward than doing an exhaustive hyperparameter sweep. This is a nice advantage over methods which require careful tuning of hyperparameters Bardes et al. (2022); Zbontar et al. (2021); Ozsoy et al. (2022).

## 7 CONCLUSION

We introduced FroSSL, a self-supervised learning method that can be seen as both sample- and dimension-contrastive. We demonstrated its effectiveness through extensive experiments on standard datasets. In particular, we discovered that FroSSL is able to achieve substantially stronger performance than alternative SSL methods when trained for a small number of epochs. To better understand why this is happening, we presented empirical results based on stepwise eigendecompositions and a comprehensive theoretical analysis. An interesting future direction of research would be to try FroSSL in combination with other SSL methods as a way of achieving faster convergence.

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
