# OpenReview forum: "FroSSL: Frobenius Norm Minimization for Self-Supervised Learning"
_ICLR.cc/2024/Conference — Submitted to ICLR 2024_

### Official Review · Reviewer_tbwD · 2023-10-26

**Soundness:** 1 poor
**Presentation:** 1 poor
**Contribution:** 1 poor
**Rating:** 3
**Confidence:** 5

**Summary:**

This paper presents FroSSL, a joint-embedding SSL method motivated by dimension-contrastive and sample-contrastive approaches. In particular, FroSSL places its central emphasis on the reduction of covariance Frobenius norms to prevent collapse and the minimization of mean-squared error to enhance augmentation invariance.
The results of this study demonstrate that FroSSL exhibits a faster convergence and learns competitive representations when compared to other SSL methods.

**Strengths:**

This paper tackles an important research problem in SSL (faster convergence).

**Weaknesses:**

Despite the mentioned strengths, the experiments and analyses presented in this paper cannot address the issue of fast convergence. Detailed weaknesses are as follows.

- The authors claim that one of FroSSL's main contributions is fast convergence. However, FroSSL is derived from rotational invariance, rather than being directly derived from fast convergence. Explanations are needed regarding the relationship between rotational invariance and fast convergence.
- The experimental results appear to lack significance. For instance, as shown in the ImageNet results of Table 4, it's evident that FroSSL converges faster than Barlow Twins only when the performance is 50% or lower. However, there are doubts about the significance of this result because it's unlikely that one would use a model with performance below 50%.
- The explanations for the figures and tables in the paper are insufficient. For example, it is unclear what the paper means by "variants" in Figure 2, and the dataset used for Figure 3 is not specified. This lack of clarity hinders the understanding of the reader.
- In Table 2, FroSSL exhibits suboptimal performance when compared with other methods.
- There seems to be a lack of experiments. (e.g., experiments on larger models where fast convergence is critical, analysis on the role of log, …)

**Questions:**

- Based on the paper, the dimension-contrastive method is characterized by its absence of negative samples, whereas the sample-contrastive method explicitly employs negative samples. Therefore, it might be inferred that dimension-contrastive and sample-contrastive approaches are inherently distinct and cannot coexist within the same framework.
However, the first contribution of this study, as corroborated by Proposition 3.3, asserts that FroSSL is simultaneously dimension-contrastive and sample-contrastive.
This apparent contradiction raises a compelling question: How can FroSSL reconcile these seemingly opposing attributes within its framework?
- What are the Figure 5.1, Figure 5.2, and Table 5.2 referring to?
- This study does not offer experimental results for the speed of convergence concerning CIFAR-10 and CIFAR-100 datasets.
- The issue of implementation effectiveness concerning the minimization of the Frobenius norm loss has been previously addressed in [1]. What sets FroSSL apart from I-VNE+ in [1], highlighting its superiority?

[1] VNE: An Effective Method for Improving Deep Representation by Manipulating Eigenvalue Distribution, CVPR 2023.

---

> ### Author Response · Authors · 2023-11-22
>
> Thank you for your review! We address your questions below
>
> **Point 1 in Weaknesses** We agree that the connection between rotational invariance and fast convergence can be made more explicitly. The connection is outlined follows. In Section 3, we choose the Frobenius norm as an operation that is invariant to unitary transformations. In Equation 9, we show that the Frobenius norm enforces the eigenvalues of the representation space to become equal. Finally, in [1], which was heavily discussed in Section 5, it is shown that learning small eigendirections leads to faster convergence. Because the Frobenius norm naturally does this, it thus can lead to fast convergence.
>
> **Point 2 in Weaknesses** We assume that Figure 4 was meant rather than Table 4. Certainly in scenarios where compute is easily available, this is perhaps not significant. However, for many groups, including us, training for 30 epochs represents a not insignificant amount of training time, roughly 1.5 days on 4 gpus. Thus the significance of our results may be highlighted in scenarios where compute is limited. As pointed out by Reviewer dD2L, experiments in the short training regime would better highlight the advantages of our method.
>
> **Point 3 in Weaknesses**  The referenced figures are explained in Section 5. The loss variants used in Figure 2 are given in Section 5.1. The Figure 3 dataset is STL10, and was specified in the first paragraph of Section 5.2.
>
> **Point 4 in Weaknesses** We copy our response to Reviewer rGwc for this point. While it is true that FroSSL only achieves competitive performance without beating baseline methods, FroSSL still retains some advantages over the superior baselines. For example, consider the CIFAR-100 column of Table 2. FroSSL is only outperformed by Barlow Twins. However, Barlow Twins has one important hyperparameter to tune ($\lambda$) while FroSSL has none. We consider this to be an important advantage of FroSSL, even if the performance is not quantitatively superior.
>
>
> **Point 5 in Weaknesses**  Due to the review period time constraints and compute resource constraints, we are unable to provide larger model experiments at this time. However, we have updated Table 2 and Table 3 to include additional results on CorInfoMax and the performance of FroSSL without logs.
>
> **Question 1** This question touches on an important property of matrices: the duality between Gram matrices and covariance matrices. In the notation of Section 3.2, these are $Z Z^T$ and $Z^T Z$, respectively. A Gram matrix represents comparisons between samples while a covariance matrix represents comparisons between dimensions. However, they share many properties such as having the same nonzero eigenvalues and having the same Frobenius norm. Methods like FroSSL that *only use these shared properties* are simultaneously dimension- and sample-contrastive. On the other hand, methods that *use nonshared properties* can only be either dimension- or sample-contastive. For example, VICReg uses the covariance matrix diagonals which is, of course, not shared by the corresponding Gram matrices.
>
>  We also note that FroSSL is not the first method to reconcile both sample- and dimension-contrastive properties. Tico has both properties too, and it was derived in a similar way to ours via Frobenius norms [2].
>
> **Question 2** Thank you for pointing out these incorrect references. These should be Figure 2, Figure 3, and Table 3, respectively. We've amended the links in our document.
>
> **Question 3** This was done because for CIFAR-10 and CIFAR-100, we cited reported results from the solo-learn library on baseline methods.
>
> **Question 4** We thank the reviewer for pointing out this great reference. We were not aware of it before and it certainly merits discussion. As discussed in Section 3.1, one interpretation of FroSSL is the maximization of Renyi $\alpha$-entropy where $\alpha=2$. The I-VNE+ method maximizes the von Neumann entropy which is a special case of Renyi $\alpha$-entropy where $\alpha=1$. One clear disadvantage of I-VNE+ is that its computation requires eigendecomposition. This can lead to slower training as batch size increases. In fact, all $\alpha\neq 2$ requires eigendecomposition. On the other hand, FroSSL avoids eigendecomposition entirely by using the Frobenius norm.
>
>
> **References**
>
> [1] James B. Simon, Maksis Knutins, Ziyin Liu, Daniel Geisz, Abraham J. Fetterman, and Joshua
> Albrecht. "On the stepwise nature of self-supervised learning."
>
> [2] Jiachen Zhu, Rafael M Moraes, Serkan Karakulak, Vlad Sobol, Alfredo Canziani, and Yann LeCun. "Tico: Transformation invariance and covariance contrast for self-supervised visual representation
> learning"

---

### Official Review · Reviewer_dD2L · 2023-10-27

**Soundness:** 2 fair
**Presentation:** 3 good
**Contribution:** 1 poor
**Rating:** 3
**Confidence:** 3

**Summary:**

The paper presents FroSSL, a novel SSL loss that is based on minimizing covariance Frobenius norms. The suggested loss is both sample- and dimension-contrastive (up to embedding normalization). Evaluated over standard datasets it is empirically shown that FroSSL experiences faster convergence and attains competitive performance using a linear probe.

**Strengths:**

[-] originality: the paper addresses an aspect which is less explored discussing SSL frameworks, the speed of convergence, which implies on the amounts of compute required for solving a given task.

[-] quality: the paper is well written, providing the reader with a clear view of diverse SSL aspects required to follow. In addition, it is evident that much work has been put into mathematical analysis as well as running experiments and benchmarks.

[-] clarity: the main ideas conveyed in this paper are clearly constructed and explained. The authors provide the necessary background, define notations and terminology, which are required to follow and understand the presented framework.

[-] significance: the paper's significance is in putting the emphasis on the convergence time of SSL objectives and suggesting an alternative that allows for faster convergence.

**Weaknesses:**

[-] contribution: the major contribution of FroSSL is presented to be its fast convergence. However, while fast convergence is a desired feature it is mostly beneficial in the case where convergence is to a competitive value or if at early epochs there is a substantial gap in performance. Judging by the provided experiments this is not the case as elaborated in the next point.

[-] overstated performance: the performance of the method is overstated judging by the provided experimental results. First, the main results (Table 2) are provided for “full” training length (1000/500 epochs), that is without utilizing the fast convergence property (and still does not outperform competing methods). Next, looking into the short training analysis over STL-10 (Table 3) already at 30 epochs SwAV is reaching nearly identical performance as FroSSL and from 50 epochs FroSSL’s performance is not in the top-3. Similarly (Sec. 6.2 and Figure 4) for ImageNet compared to Barlow Twins (BT), as highlighted by the authors, at 30 epochs the methods are already reaching similar performance, while at 100 epochs FroSSL seems to have a slight advantage over BT, it is important to note that BT is not SOTA.

[-] related work: two kernel-based methods which are very closely related to FroSSL are presented in the section for related work sec. 4.3 (“Entropy in SSL”). It will be beneficial to include these in the benchmark analysis to showcase the advantages of FroSSL in comparison to similar methods.

[-] minor: links to figures and tables are referencing respective sections. When relating to SimMER a citation is missing.

**Questions:**

[-] FroSSL’s contribution: it will be beneficial to present additional experiments where the fast convergence property stands out as significant. Alternatively, providing other properties that make FroSSL (as a dimension-contrastive and sample-contrastive method) stand out.

[-] additional baselines: as mentioned above, it will be beneficial to compare the performance of FroSSL to the presented similar methods.

---

> ### Author Response · Authors · 2023-11-22
>
> Thank you for your review! We address your concerns below
>
> **Experiments to highlight performance** Based on your comments, we agree that more experiments in the short training regime would be interesting future work.
>
> **Additional Baseline** In our updated document, we have now included comparisons to one of the baselines you mentioned, CorInfoMax. In Table 2 it is shown that FroSSL outperforms this related baseline on both CIFAR-10 and CIFAR-100. However, we were not able to achieve convergence with it on STL-10 and thus cannot compare its early epoch performance in Table 3.
>
> **Missing links and citation** Thank you for pointing out the missing citation and incorrect hyperlinks. We have fixed these in the document.

---

### Official Review · Reviewer_rGwc · 2023-10-31

**Soundness:** 3 good
**Presentation:** 3 good
**Contribution:** 2 fair
**Rating:** 5
**Confidence:** 3

**Summary:**

The authors put forward a new regularizer that is supposed to help SSL
techniques, combining both sample- and dimension-contrastive methods.

**Strengths:**

The combination follows the exposition of Garrido et al. (2023) and is
simple to understand.  The manuscript motivates the regularizer well
and has a good structure.

The faster convergence seems convincing and the explanation via
dimensional collapse is sensible.

**Weaknesses:**

The quantitative results are a bit lacking.  While the results are
always in the top 3, they never manage to beat any of the already
established methods (Table 2).

There is a mean +- std missing, aggregated over multiple runs.  I
consider this to be important because the reported accuracy numbers
are so close together in Table 2.

**Questions:**

Why do you think the FroSSL is eventually bested by other contrastive
methods over the course of training?  Is there any intuitive
explanation for it?

---

> ### Author Response · Authors · 2023-11-22
>
> Thank you for your review. We address your comments below.
>
> **Quantitative Results** While it is true that FroSSL only achieves competitive performance without beating baseline methods, FroSSL still retains some advantages over the superior baselines. For example, consider the CIFAR-100 column of Table 2. FroSSL is only outperformed by Barlow Twins. However, Barlow Twins has one important hyperparameter to tune ($\lambda$) while FroSSL has none. We consider this to be an important advantage of FroSSL, even if the performance is not quantitatively superior.
>
> **Error Bars** We agree that std would be an important addition to Table 2. Due to the time constraints of the review period, however, we are not able to include such results at this time.
>
> **Eventually bested by other methods** Based on Figure 3, one possible explanation for this is that FroSSL is able to regularize the representation space very early, while other methods take longer. However, once other methods eventually achieve this regularization, they might possibly outperform FroSSL.

---

> > ### Comment · Reviewer_rGwc · 2023-11-23
> >
> > Thank you for your response.  If the early learning dynamics are of interest to you, perhaps you could also draw a comparison to EMP-SSL, which also seems to improve the early stages of optimization (albeit with a multi-patch approach).

---

### Official Review · Reviewer_ZCeH · 2023-11-02

**Soundness:** 3 good
**Presentation:** 3 good
**Contribution:** 2 fair
**Rating:** 3
**Confidence:** 4

**Summary:**

This paper propose a new self-supervised learning algorithm. The authors give some simple analyses to show the property of this algorithm. Some experiments are conducted.

However, I think the experiments are insufficient and the motivation of their method is not convicing.

**Strengths:**

This paper propose a new self-supervised learning algorithm. The authors give some simple analyses to show the property of this algorithm. Some experiments are conducted.

**Weaknesses:**

Please see questions.

**Questions:**

* I don't understand why the objective of Barlow Twins does not push representations to be rotationally invariant, since it pushes the cross-correlation to identity.
* I think the intuition of using logarithm should be discussed further. I don't understand why it's necessary.
* Since this paper's main contribution is a new algorithm, I think its experiments are very insufficient. I highly recommend the authors to add more experiments to show the performance of their method. Besides, I think they can conduct some experiments to show the necessary of logarithm in their loss function.

---

> ### Author Response · Authors · 2023-11-22
>
> Thank you for your review. We briefly address your comments below.
>
> **Rotational Invariance** You are correct that Barlow Twins representations are invariant to rotations in the input. However, our point is that the Barlow Twins loss is not invariant to rotations in the representations. That is, only representations with an identity cross-correlation matrix achieve optimal Barlow loss. However, intuitively any rotation of identity should be equally applicable to downstream tasks. Encoding this idea into a loss function allows for a less strict optimization problem and a wider class of potential solutions.
>
> **Role of Logarithm** In our updated document, we include a comparison to FroSSL with no logarithms. This variant has obviously worse performance than FroSSL. Importantly, we do not use any tradeoff hyperparameter between the invariance and variance terms. While such a hyperparameter may improve performance without logarithms, one intuition in Section 3.1 was that the logarithm acts as a natural alternative to tradeoffs because logarithms self-regulate the influence of each of the terms in the gradient of the loss.
>
> **Additional Experiments** As discussed above, we have included experiments that show the advantage of the logarithm. We also compare to an additional baseline, CorInfoMax. We agree that more experiments would highlight the performance of FroSSL, but it requires computational resources not available to everyone.

---

### Meta-Review · Area_Chair_R9bJ · 2023-12-06

**Metareview:**

The authors propose a self-supervised learning (SSL) method based on minimizing the covariance of Frobenius norms as a means of preventing collapse while speeding up convergence, with results presented on a variety of benchmark image datasets. The reviewers agree that improving the convergence speed of SSL methods is an important problem to tackle, however, the reviewers did not find the empirical evaluation to be particularly convincing in terms of performance. The authors were not able to fully address this concern in their revisions and rebuttals.

**Justification For Why Not Higher Score:**

The reviewers are in agreement that the paper is not ready for publication, primarily based on insufficiently thorough empirical evaluation.

**Justification For Why Not Lower Score:**

N/A

---

### Decision · Program_Chairs · 2024-01-16

Reject